J Physiol 603.11 (2025) pp 3463–3477

# The impact of metformin on placental ageing in humans and mice

Grace J. Hattersley[1], Liu Yang[2], Jane L. Tarry-Adkins[3], Antonia Hufnagel[4], Kwun Kiu Wong[4] , Denise S. Fernandez-Twinn[4], Maria Chukanova[3], India G. Robinson[1], Amanda J. Drake[2], Rebecca M. Reynolds[2], Susan E. Ozanne[4,5] and Catherine E. Aiken[3,4,5]

[1]University of Cambridge School of Clinical Medicine, Cambridge, UK

[2]Centre for Cardiovascular Science, Queen's Medical Research Institute, University of Edinburgh, Edinburgh, UK

[3]Department of Obstetrics and Gynaecology, the Rosie Hospital and NIHR Cambridge Biomedical Research Centre, University of Cambridge, Cambridge, UK

[4]Institute of Metabolic Science – Metabolic Research Laboratories and MRC Metabolic Diseases Unit, Addenbrooke's Hospital, University of Cambridge, Cambridge, UK

[5]Loke Centre for Trophoblast Research, University of Cambridge, Cambridge, UK

Handling Editors: Laura Bennet & Rebecca Simmons

The peer review history is available in the Supporting Information section of this article (https://doi.org/10.1113/JP288710#support-information-section).

**The Journal of Physiology**

**Abstract figure legend** The trajectory of placental ageing following metformin treatment was assessed in three model systems: embryonic day 18.5 (E18.5) placentas obtained from obese mice treated with metformin during pregnancy, term placentas obtained from obese women randomized to treatment with metformin during pregnancy, and human primary cytotrophoblast cells isolated from term placentas at caesarean section which were treated with metformin in culture. Placental ageing was assessed at the level of the transcriptome via bulk RNA sequencing (RNAseq), qPCR, bisulphite sequencing and microarray analysis, and by assessment of tissue-level hallmarks of cellular ageing, including histology for calcification and fibrosis and telomere length analysis. Overall, we found no differences in placental ageing with metformin treatment.

**Grace Hattersley** is a fifth-year medical student at the University of Cambridge with active research involvement in placental bioinformatics. During her third-year dissertation, she used bulk RNA sequencing in a project investigating the role of an epigenetic modulator in the murine placenta. She has since been involved in research examining the effect of metformin on the developing fetus/placenta, utilizing techniques such as bulk (micro)-RNA sequencing and proteomic and lipidomic analysis. Over the remaining 18 months of her medical degree, she aims to continue optimizing her data analysis and bioinformatics skills, to meet a long-term goal of becoming a clinician scientist.

**Abstract**  Placental ageing refers to the physiological accumulation of a senescent phenotype over a healthy pregnancy. In pregnancies affected by complications such as pre-eclampsia and fetal growth restriction, placental ageing is notably accelerated and observed at an earlier gestational age. Metformin is used during pregnancy for an increasing variety of indications, including treatment of gestational diabetes, and may have a role in slowing cellular ageing. It is therefore essential to understand the potential impact of metformin on placental ageing. Placental samples ($n = 105$) were obtained from women with body mass index $\geq 30$ kg/m$^2$ and who were randomized to treatment with metformin or placebo during pregnancy. Ageing was assessed by measuring telomere length, histological examination, and using array-based technologies to investigate gene expression and methylation. Results were validated using isolated human trophoblasts treated *in vitro* with metformin, and in a complementary mouse model. There were no differences between metformin-exposed and control placentas in terms of telomere length, fibrosis or calcification. There were no differences in placental gene expression or methylation patterns by metformin status. In our mouse model, no genes classically associated with cellular ageing were differentially expressed and no senescence pathway showed evidence of enrichment with metformin treatment. There was no evidence that metformin either slows or accelerates placental ageing pathways in the complementary models that we investigated. Our findings are reassuring with regard to the safety of metformin used to treat gestational diabetes, but do not support a role for metformin in the prevention of adverse pregnancy outcomes in non-diabetic women.

(Received 9 February 2025; accepted after revision 14 May 2025; first published online 29 May 2025)

**Corresponding author** C. E. Aiken: Department of Obstetrics and Gynaecology, the Rosie Hospital and NIHR Cambridge Biomedical Research Centre, University of Cambridge, Box 223, Cambridge, CB2 0SW, UK.  Email: cema2@cam.ac.uk

## Key points

- Accelerated placental ageing, where the senescent phenotype that normally accumulates over a healthy pregnancy is observed at a premature gestational age, is associated with adverse pregnancy outcomes.
- Metformin has been proposed as an anti-ageing drug elsewhere. Therefore, metformin could alter the trajectory of placental ageing and prevent associated pregnancy complications.
- The present study incorporated human data from a randomized clinical trial and complementary models. Metformin did not impact methylation-predicted gestational age, telomere length, gene expression or histological ageing in human placentas treated *in vivo*, isolated trophoblasts treated *in vitro* or mouse models.
- Metformin neither decelerated nor accelerated placental ageing, thereby supporting its continued use in the obstetric setting, for instance in the treatment of gestational diabetes.
- Metformin cannot be recommended to prevent adverse pregnancy outcomes because we found no evidence suggesting it decelerates placental ageing. Further research is warranted to find drug therapies for this purpose.

## Introduction

Metformin is used as the first-line therapy in the treatment of gestational diabetes (GDM) in many global settings (Tarry-Adkins et al., 2021). For any drug treatment during pregnancy, there must be careful consideration of the range of possible beneficial and detrimental effects on the mother, baby and placenta (Stock & Aiken, 2023). With respect to metformin, which has been suggested as an anti-ageing therapy in other contexts (Kulkarni et al., 2020), there is particular rationale to examine closely whether there is any impact of treatment on placental ageing.

The placenta naturally ages throughout a healthy pregnancy (Cox & Redman, 2017). From the earliest stages of formation of the syncytiotrophoblast by cell fusion, there is concomitant activation of senescence-associated molecular pathways and expression of ageing markers,

including p16, p21 and p53 (Chuprin et al., 2013). Human syncytiotrophoblasts in healthy pregnancy show progressive evidence of DNA oxidation (e.g. 8-hydroxy-deoxyguanosine accumulation) as gestation increases (Maiti et al., 2017). This accumulation of senescent markers is accelerated in comparison to other tissues, given that the human placenta exists for a temporary 9 month period and has a condensed ageing trajectory. Well-defined epigenetic changes also occur as the human placenta ages over gestation, and these have been collated into CpG signatures or 'placental clocks' that can be used to accurately predict gestational age (Lee et al., 2019).

Accelerated placental ageing is observed in adverse pregnancy outcomes, including stillbirth, spontaneous preterm birth and fetal growth restriction (Sultana et al., 2018). These complications are urgent health priorities worldwide, affecting over 15% of pregnancies globally (Chawanpaiboon et al., 2019; Hug et al., 2021; Lu et al., 2017). There is also acceleration of the physiological placental ageing process in response to additional stressors, such as maternal obesity (Martens et al., 2016; Tao et al., 2023), and maternal obesity is similarly associated with an increased prevalence of serious pregnancy complications (Catalano & Ehrenberg, 2006) Currently, few effective strategies are available to reduce the incidence of adverse pregnancy outcomes related to placental ageing (Stock & Aiken, 2023). Therefore, a thorough evaluation of any intervention that might alter the trajectory of placental ageing is of high clinical importance.

Several mechanisms of action attributed to metformin are associated with slowing the ageing process, for instance inhibition of mitochondrial complex I and activation of 5′ AMP-activated protein kinase (AMPK) and Sirtuin1 (SIRT1). Clinical trials are currently underway in different contexts to evaluate metformin's efficacy in reducing ageing-associated tissue atrophy and its ability to increase tissue markers of longevity (Khan et al., 2023). As a drug already commonly used in pregnancy, particularly in the treatment of GDM and pre-existing diabetes mellitus (Tarry-Adkins et al., 2021), metformin could be a potential candidate for slowing placental ageing. We have previously shown reduced oxidative stress and lower complex I activity in cultured human trophoblasts in response to metformin at clinically relevant doses (Tarry-Adkins et al., 2022). Thus, there is a strong rationale to fully investigate whether metformin may have an anti-ageing effect in the human placenta.

## Methods

### Ethical approval

The EMPOWaR trial was approved by the research ethics committee (REC 10/MRE00/12) and the Medicines and Healthcare products Regulatory Agency (EudraCT number 2009-017134-47), and all participants provided written informed consent. The human studies included in this paper conformed to the standards of the *Declaration of Helsinki*, except for registration in a database. The University of Cambridge Animal Welfare Ethical Review Body (AWERB) reviewed and approved all animal work, which was performed in accordance with the United Kingdom Animals (Scientific Procedures) Act of 1986 and under the animal project licence P5FDF0206 issued by the UK Home Office. All animal work complies with the standards stated for the *Journal of Physiology* (Grundy, 2015).

### Model systems

**EMPOWaR trial (Fig. 1A).** Pregnant women aged >16 years with body mass index (BMI) $\geq$ 30 kg/m$^2$ were recruited between 12 and 16 weeks' gestation to participate in a randomized controlled trial (EMPOWaR) (Chiswick, Reynolds, Denison, Whyte, et al., 2015; Chiswick, Reynolds, Denison, Drake, et al., 2015, Chiswick et al., 2016). Included women were randomly assigned to receive 500–2500 mg oral metformin or matched placebo tablets from 12–16 weeks' gestation until delivery. Treatment with metformin was commenced at a dose of 500 mg once daily in Week 1. The daily dosage was increased by 500 mg each week over a 5 week period, until the highest tolerable dose or 2500 mg was reached. If side effects occurred, participants were advised to revert to the previous week's dose for 1 week, before re-attempting any increase. Adjustments according to the discretion of the local medical team were permitted, provided the daily dose did not exceed 2500 mg in three divided doses. Exclusion criteria were: non-white ethnicity, pre-existing diabetes, GDM in a prior pregnancy or in the index pregnancy preceding randomization, systemic disease at trial entry currently requiring medical management or systemic corticosteroids within the past 3 months, previous delivery of a baby with birth weight smaller than the third percentile, history of pre-eclampsia in a previous pregnancy that necessitated delivery before 32 weeks' gestation, known hypersensitivity to metformin or any of its components, diagnosed liver or renal failure, acute medical conditions at trial entry with the potential to impact renal function or to cause tissue hypoxia, breastfeeding, and multiple pregnancy. Baseline demographics did not differ between the metformin and control groups. Placental samples were collected at birth ($n = 103$). DNA and RNA were extracted from tissue using AllPrep DNA/RNA/protein Minikits according to the manufacturer's instructions (Qiagen, Crawley, UK). Stored placental samples were provided by the Edinburgh Reproductive Tissue Biobank (ERTBB; REC20/ES/0061).

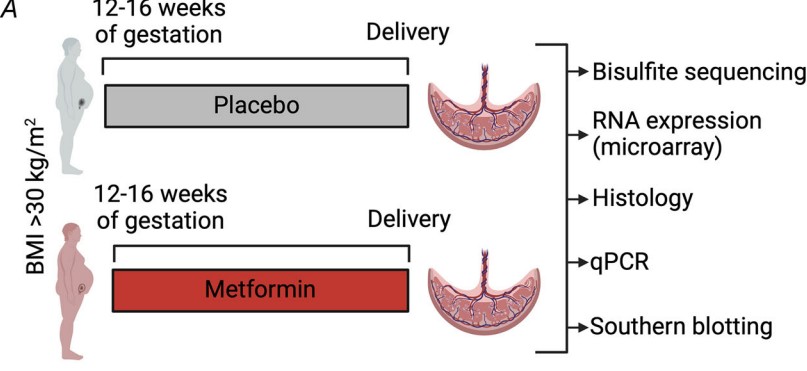

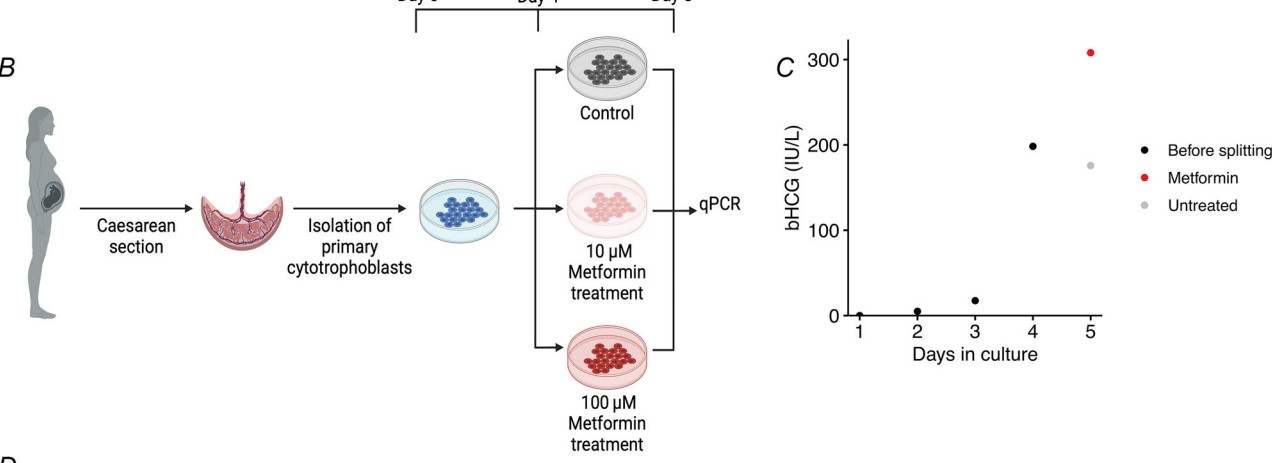

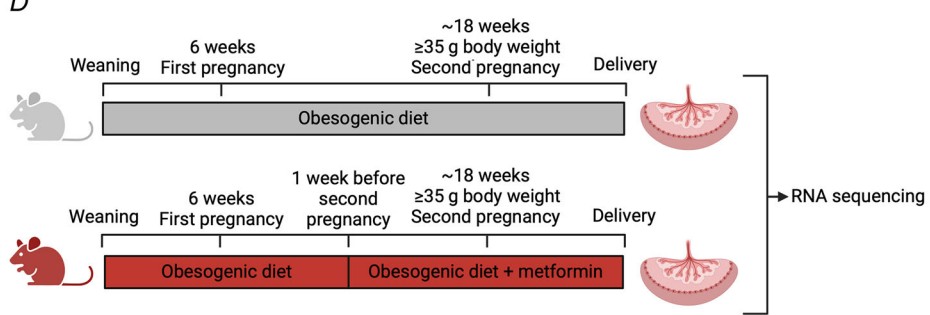

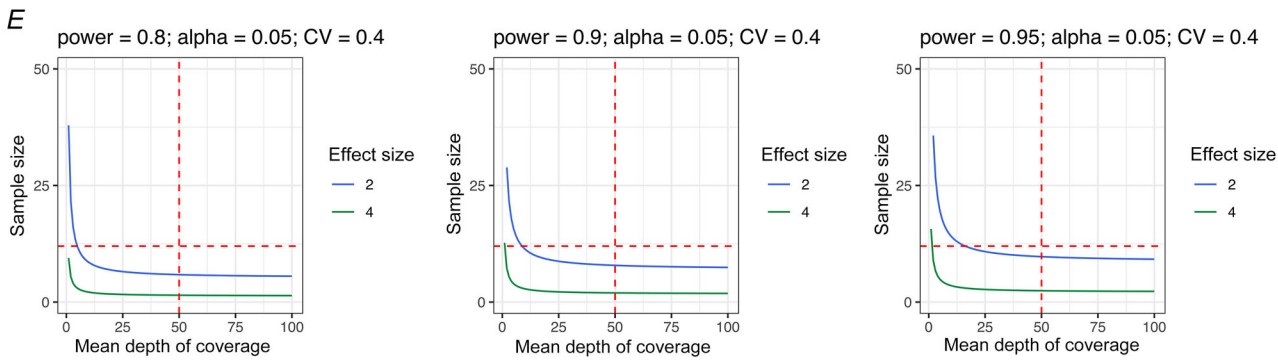

**Figure 1. Model systems and methods**
*A*, EMPOWaR trial. Placentas were collected at term from obese pregnancies treated with placebo or metformin
and processed. *B*, human trophoblast model. Primary cytotrophoblasts were isolated from placentas delivered by
caesarean section at term for treatment with placebo or metformin and processing for qPCR. *C*, beta-hCG rise in
cultured human cytotrophoblasts over time. Representative data from *n* = 1 placenta. The cells are treated with

metformin on Day 4, as described in the Methods. *D*, mouse model. Obese mice were treated with metformin or placebo during their secondary pregnancy. Placentas collected at term were processed for bulk RNAseq. *E*, power calculations for murine bulk RNAseq. The mean coefficient of variation (CV) for the RNAseq data was 0.4, and a $\log_2$-fold change of 1, corresponding to an effect size of 2 (blue line), was deemed significant. The horizontal red dashed line corresponds to a sample size of 12 and the vertical red dashed line corresponds to a mean depth of coverage of 50, which was exceeded in all samples. Power ($1 - \beta$) and alpha values are reported in the plot titles. [Colour figure can be viewed at wileyonlinelibrary.com]

## Placental collections, trophoblast isolation and culture (Fig. 1*B*).

Placentas were donated by women undergoing elective Caesarean section at term without prior onset of labour who provided informed consent (Tarry-Adkins et al., 2023). Exclusion criteria were multiple pregnancy, presence of a major fetal anomaly, diagnosis of severe pre-eclampsia or any form of diabetes, gestational age <37 weeks and use of metformin during the pregnancy. Primary cytotrophoblasts were isolated from these freshly collected placentas using DNase digestion and centrifugation according to protocols described in detail elsewhere (Tarry-Adkins et al., 2022). Isolated primary cytotrophoblasts were plated in culture medium at various densities to differentiate into syncytiotrophoblasts at 37°C under 5% $CO_2$ and atmospheric oxygen (Tarry-Adkins et al., 2023). After 96 h of culture, cells were treated with metformin at concentrations of 10 and 100 μM. After 120 h of culture, cells were harvested. Whilst in culture, the medium was changed every 24 h. Beta-human chorionic gonadotropin (beta-hCG) levels continued to rise throughout the time in culture (Fig. 1*C*). Placental collections were obtained via the biobanks of the Centre for Trophoblast Research (REC17/EE/0151) and from the Cambridge Blood and Stem Cell Biobank (REC18/EE/0199).

## Mouse model of metformin treatment in obese pregnancy (Fig. 1*D*).

Wild-type C57BL/6J mice obtained from Charles River Laboratories (Wilmington, MA, USA) were bred in-house. Mice were single-housed and kept in individually ventilated cages with wood chip bedding and free access to food, water and environmental enrichment (nesting material and a tunnel) in a 12 h light/dark cycle. Female mice were *ad libitum* fed a well-established obesogenic diet (no. 824053, Special Dietary Services) (Fernandez-Twinn et al., 2012) supplemented with condensed milk (no. 12029969, Nestlé) from weaning (3 weeks of age). A first pregnancy was commenced at 6 weeks of age, and mating for the second pregnancy was commenced when the mice achieved a body weight of ≥35 g (~18 weeks of age). The day of the plug was considered embryonic day 0.5 (E0.5) and fetuses were killed at E18.5 by cardiac puncture under 2% isoflurane anaesthesia. Death was confirmed by cervical dislocation. Mice were randomized to receive metformin (0215169-CF, MP Biomedicals, Irvine, CA, USA) in the condensed milk 1 week before mating and throughout the subsequent pregnancy to a dose of 300 mg/kg/day (equivalent to 1.7 g in a 70 kg man) (Salomäki et al., 2014). We calculated that a sample size of 12 per group would be sufficiently powered to detect differential gene expression using a $\log_2$-fold change of 1, a mean coefficient of variation of 0.4 and a mean depth of coverage of 50 (Fig. 1*E*).

## Experimental techniques

**Telomere length analysis.** High-molecular-weight DNA was extracted using the DNeasy Blood and Tissue kit (Qiagen, Hilden, Germany) according to the manufacturer's instructions. DNA quantity and purity were determined using a NanoDrop spectrophotometer (NanoDrop Technologies, Wilmington, DE, USA). Southern blotting was performed as fully described elsewhere (Aiken et al., 2019). Telomere signals were analysed using Adobe Photoshop (Adobe Systems Inc., San Jose, CA, USA) and Alpha-Ease software (Alpha Innotech, San Leandro, CA, USA). Telomere length was divided into categories described previously (1.34.2, 4.2–8.6, 8.6–21.2, 21.2–48.5 kilo-bases) (Tarry-Adkins et al., 2006).

**qPCR.** RNA was extracted from trophoblasts, quantified using a NanoDrop spectrophotometer (Thermo Fisher Scientific) and checked for integrity by running an agarose gel to check for the presence of the 18S and 28S ribosomal RNA bands, then synthesized to cDNA as detailed previously (Tarry-Adkins et al., 2009). Gene expression was determined by RT-PCR using custom-designed primers (Merck, Darmstadt, Germany), which were designed to fall within one exon and run against human genomic DNA (gDNA) standard curves to determine copy numbers. SYBR Green PCR master mix (Applied Biosystems, Waltham, MA, USA) was used as previously described (Tarry-Adkins et al., 2009). Equal efficiency of the reverse transcription of RNA from all groups was confirmed through quantification of expression of the housekeeping gene *B2M*, the expression of which did not differ between groups. For placental samples, cDNA from RNA samples was used in Taqman expression assays (Life Technologies, UK), with *YWHAZ* as an endogenous control. Gene expression was calculated and normalized to a plate-control sample.

**RNA sequencing and analysis.** In total, 35–45 mg of placental tissue was isolated from 12 control and 12 metformin-treated placentas. RNA was extracted using the miRNeasy micro kit (Qiagen), with DNA contamination removed using the DNAse kit (Qiagen). Both steps were performed according to the manufacturer's instructions. RNA quality and concentration were assessed by NanoDrop analysis. An RNA integrity number (RIN) of >8 for all samples was confirmed by analysis with the Agilent Bioanalyser 2100 system (Agilent RNA 6000 Nano Kit). In total, 400 ng of total RNA was used for library construction (TruSeq Stranded mRNA Library Pre Kit; Illumina, San Diego, CA, USA). Indexed libraries were normalized, pooled and sequenced on a NovaSeq 6000 (50 bp paired-end).

**Microarrays.** Gene expression arrays (Illumina HT12v4) were performed on cRNA samples (Illumina Total Prep cRNA kit, Life Technologies, UK). Further analysis of array results used KEGG pathway IDs, normalized values for gene expression or DNA methylation, and gene function (www.genecard.org) to identify relevant genes. The arrays were QC analysed using the array Quality Metrics package in Bioconductor. Arrays were scored (outliers identified) based on MA plot, boxplot, heatmap and manual inspection. An array was classified as an outlier if it failed two or more parameters.

**Methylation arrays.** DNA methylation arrays (Illumina Infinium HumanMethylation450 BeadChips) were performed on bisulphite-converted DNA (EZ DNA Methylation kit, Zymo Research, UK). Raw data from the arrays were pre-processed by using the 'minfi' R package. Samples ($n = 2$) that failed quality control (separate and lower median intensity in both methylated and unmethylated sites) were excluded. A normal-exponential out-of-band (noob) method was used for background noise and dye bias normalization (Triche et al., 2013). A final number of 485,512 CpG sites and 103 samples were included for further analysis.

**Histology.** Full-thickness placental samples ($\geq$10 mm in diameter) were immersion-fixed in 10% neutral-buffered formalin and processed. Alizarin Red staining was performed to analyse placental calcification. Sections were dewaxed into 95% alcohol, air-dried and incubated in Alizarin Red (Sigma–Aldrich, Cat. #A-5533, St Louis, MO, USA). Slides were counterstained with fast green stain (0.05% Fast Green FCF; Sigma–Aldrich, Cat. #F-7252). Slides were washed, dehydrated, cleared and mounted in synthetic resin. For fibrosis, sections were dewaxed, then stained with anti-fibrinogen (Abcam 34269, Cambridge, MA, USA), followed by incubation in antigen retrieval buffer (Vector H-3300, Burlingame, CA, USA). Washing

(Dako S3006, Carpinteria, CA, USA) and blocking (Vector, SP-6000/SP-5030) were performed. Sections were incubated with primary anti-fibrinogen antibody (Abcam 34269; 1:1000) at 4°C overnight. Sections were washed, then secondary antibody (goat anti-rabbit, 1:1000) was added. ImmPRESS HRP (goat anti-rabbit Vector, MP-7451) and ImmPACT DAB (Vector, SK-4105) were added, and then counterstaining with Hematoxylin QS (Vector H-3404) was performed. Slides were immersed in xylene and then mounted. Images were acquired on a slide scanner (Zeiss Axioscan 7, Oberkochen, Germany) and then analysed automatically (HALO Image Analysis Platform 3.6.4132). Staining was expressed as a percentage of the whole section.

## Statistical analysis

Descriptive statistics are displayed as mean ± standard deviation (SD) or median ± interquartile range (IQR) according to data distributions, and *P*-values were calculated using Student's *t* tests (two-sided), Mann–Whitney tests or one-way ANOVA. Multivariable regression models were generated to assess the impact of treatment group on outcomes of interest and adjusted as appropriate for maternal characteristics, including BMI, where applicable. Correction for multiple hypothesis testing was performed by applying a correction for false discovery rate (FDR) where necessary.

For RNA sequencing (RNAseq) data, STAR was used to map reads to the GRCm38 genome. Ensembl IDs were converted into gene names using annotations provided by BioMart. Genes with a total count across all samples <10 were excluded. Differential expression analysis was performed using DESeq2, with the variance stabilizing transformation (VST) function to normalize count data. Gene Set Enrichment Analysis (GSEA) was conducted using the fgsea package, with the rank metric calculated:

$$\text{Rank metric} = \left|\log \text{fold change}\right| * -\log_{10}\left(P - \text{value}\right)$$

Gene ontology, REACTOME and the FRIDMAN_SENESCENCE_UP gene sets were downloaded from the Molecular Signature Database (MSigDB).

Statistical analysis was performed using R for Statistical Computing v4.4.1. Figures were plotted using the ggplot2, pheatmap and EnhancedVolcano packages.

## Results

### Effect of metformin on age-associated placental gene methylation in human placentas from pregnancies affected by obesity

Methylation patterns in the human placenta vary predictably with gestational age, giving rise to the concept of

a 'placental clock' (Lee et al., 2019; Mayne et al., 2017). We used an array-based method to assess whether the methylome of term placentas obtained in the EMPOWaR trial from women with obesity randomized to receive metformin during pregnancy ($n = 51$) displayed accelerated ageing compared to placentas from gestational age-matched women randomized to placebo ($n = 52$) (Chiswick, Reynolds, Denison, Drake, et al., 2015).

Differential methylation at 2873 CpG sites was seen at raw *P*-values of <0.01, but there were no FDR-adjusted differences in DNA methylation between the gestational age-matched metformin- and placebo-treated groups.

We used three previously described CpG signatures ('placental clocks') that reflect placental ageing to assess the correlation between predicted and true gestational age (Fig. 2*A*) (Lee et al., 2019). We selected the placental clock that gave the best correlation between predicted and true gestational age for further analysis (refined robust placental clock; mean absolute error = $-1.4$ weeks, $R^2 = 0.63$, $P < 0.001$). In our true gestational

age-matched cohort, there were no differences in the gestational age predicted by the placental clock model for metformin-treated *versus* control placentas (Fig. 2*B*; metformin: $38.38 \pm 1.50$ weeks *vs.* control: $38.24 \pm 1.18$ weeks, $P = 0.61$). Overall, there was therefore no difference in the rate of ageing in the placentas of women randomized to metformin compared to control as part of the EMPOWaR trial based on predictable differences in methylation status.

## Effect of metformin on expression of ageing-associated genes in human placenta

We compared gene expression in whole placentas from gestational age-matched women with BMI > 30 kg/m$^2$ randomized to metformin ($n = 52$) or placebo ($n = 53$) using an array-based methodology (Chiswick, Reynolds, Denison, Drake, et al., 2015). Differential expression of 146 gene probes was seen at raw *P*-values of <0.01, but there were no FDR-adjusted differences in

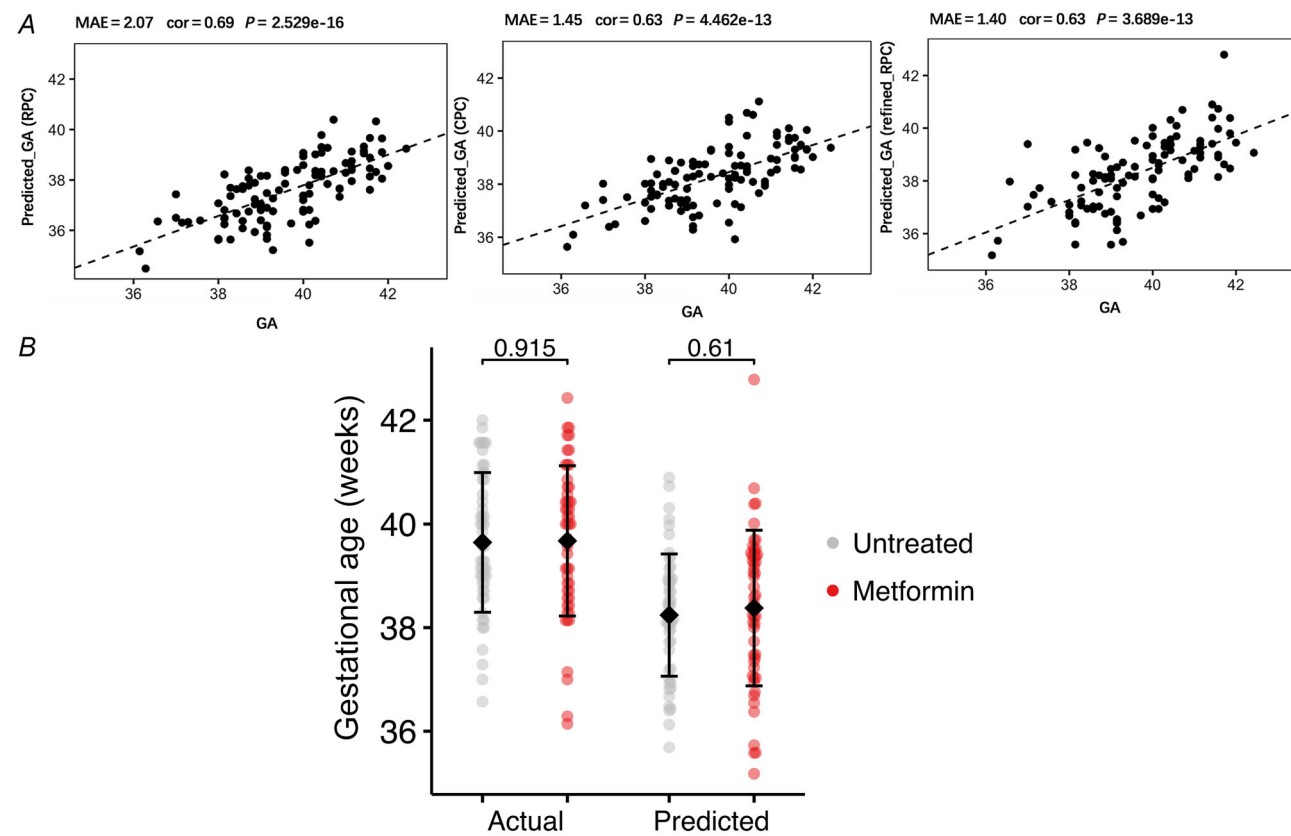

**Figure 2. Correlation between predicted gestational age based on epigenetic signatures and true gestational age in metformin-treated and control placentas ($n = 103$)**
*A*, true gestational age against gestational age predicted by the RPC, CPC and refined RPC regression models in control and metformin-treated placentas ($n = 103$) (RPC = robust placental clock; CPC = control placental clock; Refined_RPC = refined robust placental clock; GA = gestational age; Cor = correlation coefficient; MAE = mean absolute error). *B*, differences between true and predicted gestational age in metformin-treated ($n = 51$) *versus* control ($n = 52$) placentas. Mean and standard deviations are shown. Pairwise comparisons were calculated using Student's *t* tests. [Colour figure can be viewed at wileyonlinelibrary.com]

placental mRNA levels between the metformin- and placebo-treated groups.

Fourteen genes differentially expressed at raw *P*-values of <0.01 with known relevant ageing, mitochondrial or metabolic functions were selected for validation by qPCR (Table 1). Two of these genes, *PPARGC1A* and *ERBB4*, had reduced mRNA levels with metformin treatment after adjusting for potential confounders (*PPARGC1A*: beta coefficient −0.217, *P* = 0.048; *ERBB4*: beta coefficient −0.344, *P* = 0.002, respectively).

To test whether the mRNA expression of *PPARGC1A* or *ERBB4* was specifically reduced in trophoblasts, rather than whole placental lysates, we isolated human trophoblasts from fresh placentas delivered at elective Caesarean section at full term. Trophoblasts from each placenta were split and cultured *in vitro* with either 100 μM metformin, 10 μM metformin or vehicle (*n* = 15 per group). Consistent with the whole placenta data, expression of *PPARGC1A* RNA in isolated trophoblasts was reduced with increasing doses of metformin from mean *z*-score 0.36 in control trophoblasts to 0.24 in 10 μM metformin to −0.60 in 100 μM metformin (ANOVA *P* < 0.001; Fig. 3*A*). Conversely, the expression of *ERBB4* RNA increased with increasing doses of metformin from a mean *z*-score of −0.25 in control trophoblasts to −0.13 in 10 μM metformin and to 0.38 in 100 μM metformin (ANOVA *P* = 0.001; Fig. 3*B*).

In addition to the array-based approach, we also took a 'candidate gene' approach to detecting ageing-associated differences in gene expression in the placentas of gestational age-matched women who participated in the EMPOWaR trial with BMI > 30 kg/m$^2$ randomized to metformin or placebo (*n* = 24 per group). RNA expression of a panel of 10 key ageing and oxidative stress response-associated genes was measured using qPCR, with no difference in gene expression between the control and metformin-treated placentas (Fig. 3*C*).

## Effect of metformin on ageing gene expression in mouse placentas from obese mothers

To further explore whether metformin affects ageing-related pathways in the placental transcriptome across species, we performed bulk RNAseq of placentas from metformin-treated *versus* control obese pregnant mice obtained on E18.5 of gestation (*n* = 12 per group). Principal component analysis showed no clear clustering by metformin treatment status (Fig. 4*A*), and there was no significant difference in the principal component loadings for untreated controls *versus* metformin-treated placentas (Fig. 4*B*), indicating only weak effects of metformin on the global placental transcriptome. Only three differentially expressed genes were identified in metformin-treated *versus* control placentas, of

which just *Prl4a1*, a non-classical prolactin, showed a log-fold change (LFC) of greater than 0.5 (Fig. 4*C*; Table 2).

To further elucidate the effect of metformin on placental ageing at the transcriptome level, we performed gene set enrichment analysis using six cellular senescence pathways obtained from the Molecular Signatures Database (MSigDB). None of these pathways showed differential expression in metformin-treated compared to control obese mice placentas (Fig. 4*D*). A senescence pathway defined experimentally from transcripts universally differentially expressed across a plethora of senescence models was also tested (Casella et al., 2019). Neither the upregulated nor downregulated transcripts showed differential expression in the metformin-treated placentas compared to controls (Fig. 4*E*).

We also looked for evidence of metformin treatment affecting gene pathways linked to the clinical sequelae of placental ageing. Placental ageing is implicated in the aetiology of preterm birth (Sultana et al., 2018). Therefore, pathways enriched in the placental transcriptomes from spontaneous preterm compared to term births may reflect accelerated placental ageing (Akram et al., 2022). There was no enrichment of these pre-term birth-associated genes in metformin-treated compared to control placentas (Fig. 4*F*).

## Effect of metformin on human placental telomere length

Given that we found little evidence of metformin treatment affecting placental ageing at the transcriptomic level, we next assessed whether there might nonetheless be downstream impacts of metformin treatment. Specifically, we investigated tissue-level hallmarks of placental ageing, including telomere length, calcification and fibrosis (Ferrari et al., 2016; Ohmaru-Nakanishi et al., 2018; Poggi et al., 2001).

We measured telomere length in whole placentas from women with BMI > 30 kg/m$^2$ who participated in the EMPOWaR trial and were randomized to receive metformin (*n* = 24) or placebo (*n* = 23) during pregnancy (Chiswick, Reynolds, Denison, Whyte, et al., 2015). Average telomere lengths were not different in metformin-treated compared to placebo treated whole placental tissue (Fig. 5*A*). There was no difference between metformin-treated *versus* control placentas in terms of either the ratio of longest-to-shortest telomeres (Fig. 5*B*) or the overall distribution of telomere lengths (Fig. 5*C*). As expected, the percentage of very short (1.3–4.2 kb) telomeres increased with gestational age but was not different between the metformin-treated and control groups (Fig. 5*C*).

Table 1. Results of linear regression models of the relative expression (qPCR) of each gene in human placentas treated with metformin ($n = 52$) or placebo ($n = 53$) during pregnancy. Models were adjusted for treatment, labour, maternal age (years), sex of baby, birth weight (g, adjusted for gestational age) and GDM. Significant predictors are highlighted in bold

| Gene | Overall model | Treatment group | | Labour | | Maternal age | | Sex of baby | | Birth weight | | GDM | |
|---|---|---|---|---|---|---|---|---|---|---|---|---|---|
| | ANOVA P-value | Beta value | P-value | Beta value | P-value | Beta value | P-value | Beta value | P-value | Beta value | P-value | Beta value | P-value |
| CDC42 | 0.672 | −0.035 | 0.763 | −0.086 | 0.469 | 0.028 | 0.809 | 0.046 | 0.688 | −0.202 | 0.098 | 0.039 | 0.732 |
| ERBB4 | **0.015** | **−0.344** | **0.002** | **0.237** | **0.034** | 0.096 | 0.371 | 0.103 | 0.332 | **0.301** | **0.009** | −0.031 | 0.767 |
| INPP5A | **0.013** | −0.113 | 0.297 | **0.391** | **0.001** | −0.123 | 0.250 | −0.035 | 0.743 | **0.259** | **0.023** | 0.047 | 0.654 |
| MAP2K2 | 0.901 | −0.139 | 0.242 | 0.090 | 0.457 | −0.060 | 0.608 | 0.020 | 0.863 | 0.081 | 0.511 | −0.027 | 0.814 |
| PAK1 | **0.012** | −0.074 | 0.492 | **−0.301** | **0.007** | 0.123 | 0.250 | **−0.221** | **0.040** | **0.279** | **0.014** | −0.103 | 0.326 |
| PIK3R1 | 0.228 | 0.104 | 0.362 | 0.117 | 0.313 | −0.046 | 0.682 | −0.001 | 0.995 | −0.190 | 0.110 | −0.130 | 0.239 |
| PIK3R5 | **0.023** | −0.021 | 0.848 | **0.394** | **0.001** | −0.099 | 0.359 | 0.136 | 0.205 | 0.141 | 0.215 | 0.057 | 0.593 |
| PPARGC1A | **0.015** | **−0.217** | **0.048** | **0.293** | **0.009** | −0.210 | 0.051 | −0.082 | 0.443 | 0.196 | 0.084 | −0.084 | 0.425 |
| PRKCB | 0.525 | 0.009 | 0.941 | 0.169 | 0.155 | 0.008 | 0.944 | 0.156 | 0.172 | 0.191 | 0.114 | −0.029 | 0.794 |
| SLC22A3 | **0.024** | −0.160 | 0.146 | **0.367** | **0.001** | −0.059 | 0.582 | 0.147 | 0.172 | **0.285** | **0.014** | 0.017 | 0.875 |
| SLC25A4 | **0.018** | −0.002 | 0.988 | **0.415** | **<0.001** | −0.047 | 0.658 | −0.024 | 0.819 | 0.047 | 0.679 | −0.060 | 0.567 |
| SLC25A35 | 0.166 | −0.250 | 0.029* | −0.011 | 0.922 | 0.137 | 0.220 | 0.042 | 0.705 | 0.197 | 0.096 | −0.061 | 0.578 |
| TIMM10 | **0.004** | −0.168 | 0.115 | **0.428** | **<0.001** | −0.045 | 0.670 | −0.040 | 0.703 | **0.311** | **0.006** | −0.025 | 0.806 |
| TMEM126B | 0.052 | −0.102 | 0.360 | **0.373** | **0.001** | −0.093 | 0.394 | 0.112 | 0.304 | 0.120 | 0.296 | −0.030 | 0.778 |

* Although expression of *SLC25A35* was associated with treatment, the overall model was not significant ($P = 0.166$).

## Effect of metformin on placental calcification and fibrosis in human placentas from pregnancies affected by obesity

Fixed sections of metformin-exposed ($n$ = 24) and control ($n$ = 24) placentas obtained from women who participated in the EMPOWaR trial were stained to detect calcium deposits and fibrosis. There was no difference between the extent of either calcification (control calcification: median = 0.010%, IQR = 0.31; metformin calcification: median = 0.015%, IQR = 0.17; $P$ = 0.88; Fig. 6A) or fibrosis (control fibrin: median = 22%,

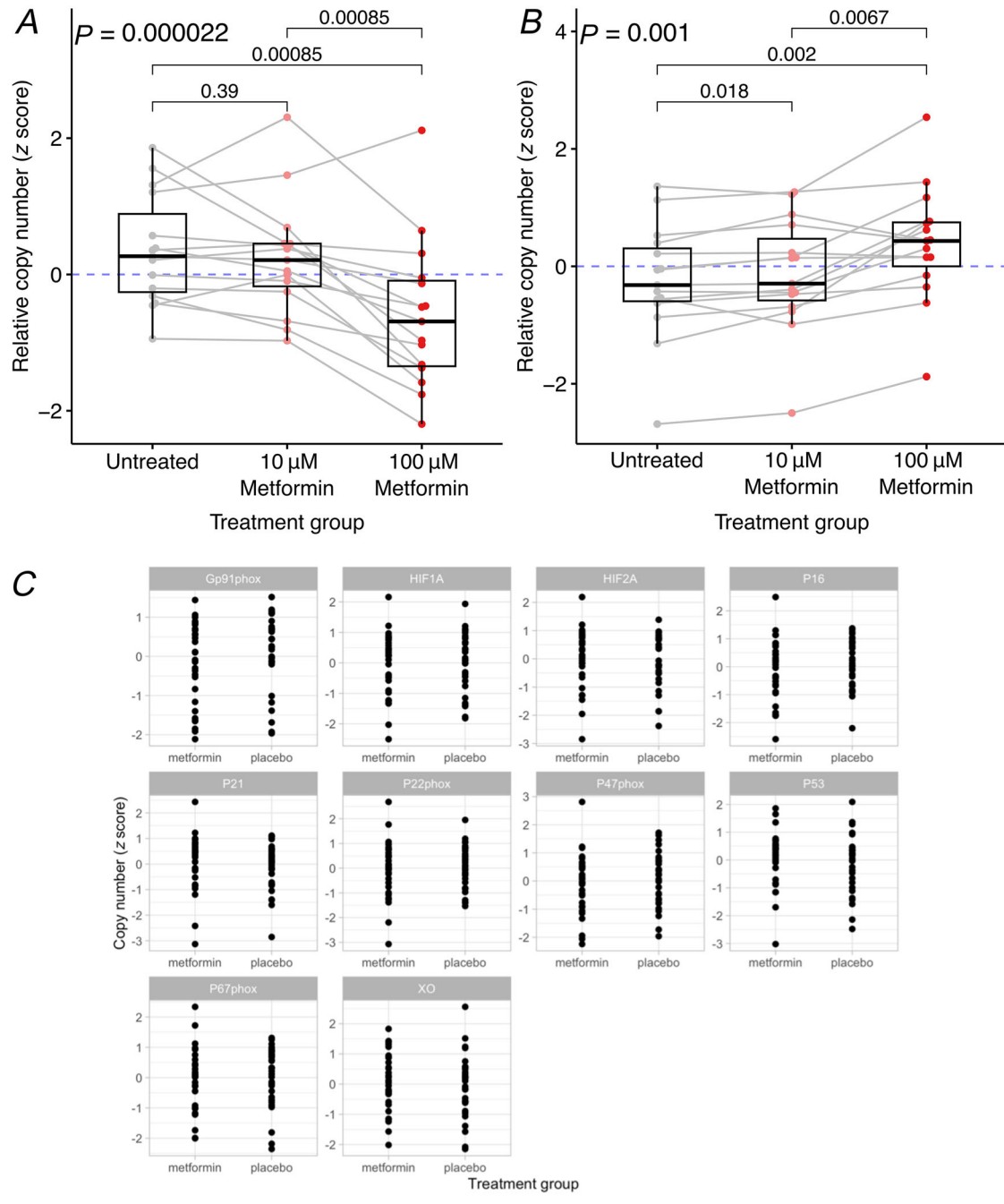

**Figure 3. Expression of ageing associated genes in metformin-treated and control placentas**
*A* and *B*, expression on qPCR of *PPARGC1A* (*A*) and *ERBB* (*B*) genes in isolated human trophoblasts following placebo ($n$ = 15), 10 μM metformin ($n$ = 15) or 100 μM ($n$ = 15) treatment. Medians and IQR are shown. One-way ANOVA and Mann–Whitney *U* test *P*-values are indicated. *C*, expression on qPCR of 10 key ageing and oxidative stress response-associated genes in whole human placentas treated with placebo ($n$ = 24) or metformin ($n$ = 24) during pregnancy. [Colour figure can be viewed at wileyonlinelibrary.com]

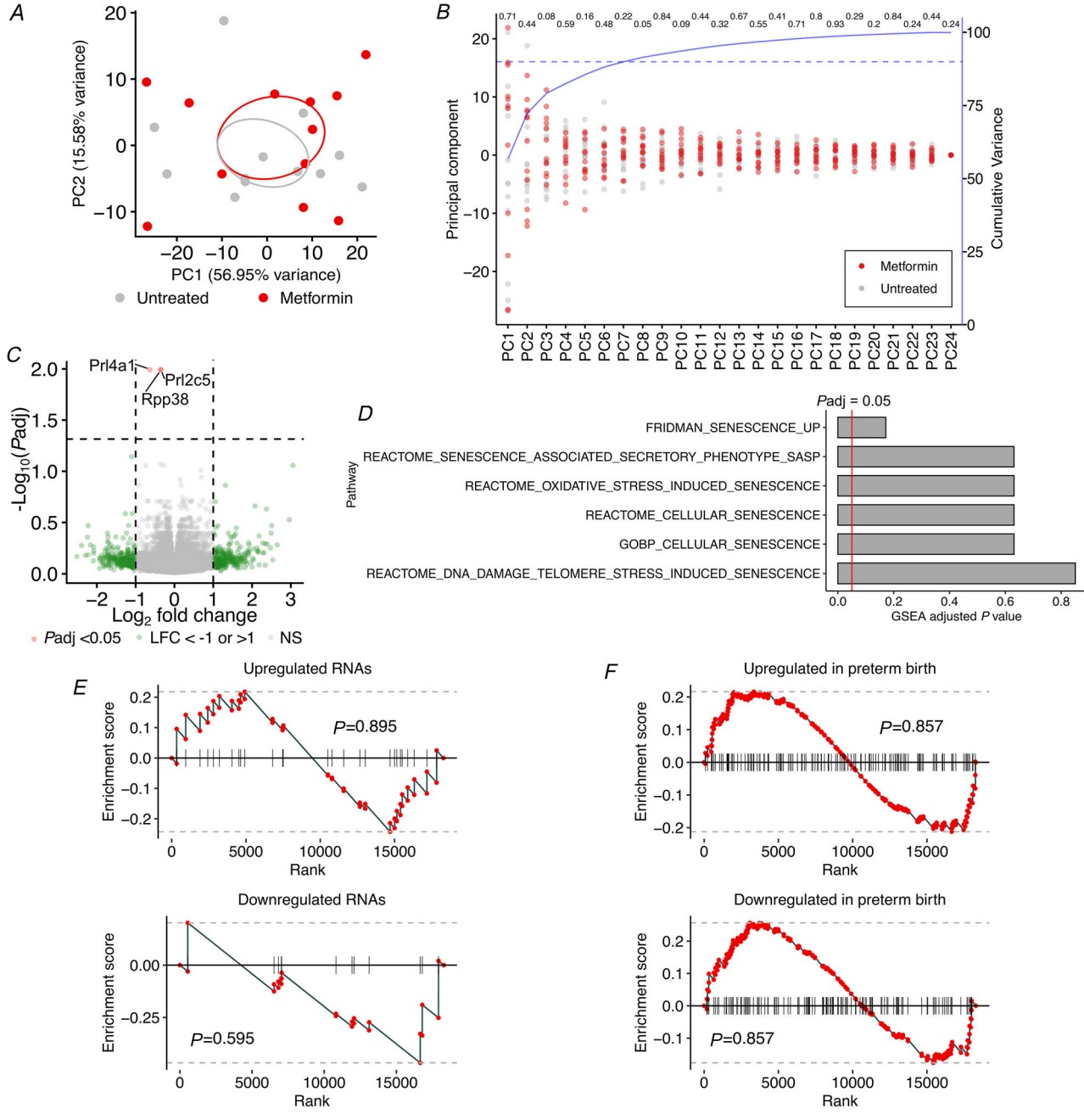

**Figure 4. Gene expression in metformin-treated mice (*n* = 12) compared to control obese mice (*n* = 12)**
*A*, principal component (PC) analysis of RNAseq data of obese placentas. Grey = control obese mice; red = metformin-treated obese mice. *B*, principal component analysis of RNAseq data of obese placentas. Right *y*-axis (blue) shows the cumulative variance accounted for by each principal component. The dashed line indicates 90% cumulative variance. *P*-values compare principal component loadings for untreated controls *versus* metformin-treated placentas and were calculated by Mann–Whitney *U* test. Grey = control obese mice; red = metformin-treated obese mice. *C*, volcano plot demonstrating differentially expressed genes in metformin-treated *versus* control obese mice placentas. Adjusted *P*-value cut-off = 0.05. Absolute $\log_2$ fold change cut-off = 1. LFC = $\log_2$ fold change; *P*adj = adjusted *P*-value; NS = not significant (*P* ≥ 0.05, absolute LFC < 1). *D*, adjusted *P*-values from gene set enrichment analysis (GSEA) of selected cellular ageing pathways obtained from the Molecular Signatures Database (MSigDB). *E*, GSEA of gene signatures upregulated (top) and downregulated (bottom) in senescence in metformin compared to control obese murine placentas. Gene signatures derived from Casella et al. (2019), GEO: GSE130727. *F*, GSEA of gene signatures upregulated (top) and downregulated (bottom) in preterm birth in metformin compared to control obese murine placentas. Gene signatures derived from Akram et al. (2022), GEO: GSE211927. [Colour figure can be viewed at wileyonlinelibrary.com]

**Table 2. Log fold change and *P*-values of differentially expressed genes in metformin-treated (*n* = 12) and control (*n* = 12) murine placentas. Results in bold indicate significance (*P* < 0.05)**

|        | Log₂-fold change | *P*-value | Adjusted *P*-value |
|--------|------------------|-----------|--------------------|
| PRL2C5 | −0.348           | **0.00000106** | **0.0105**     |
| PRL4A1 | −0.633           | **0.00000170** | **0.0105**     |
| RPP38  | −0.361           | **0.00000172** | **0.0105**     |

IQR = 14.14; metformin fibrin: median = 18%, IQR = 11.30; *P* = 0.25; Fig. 6*B*). Similarly, multivariable linear regression adjusting for maternal age at conception, length of gestation in days, delivery method (vaginal or caesarean section), labour type (spontaneous or induced/caesarean section), offspring sex and birth weight did not reveal any difference in calcification (*P* = 0.55) or fibrosis (*P* = 0.24) between treatment groups.

## Discussion

Accelerated placental ageing is commonly observed in complicated pregnancies, including those affected by pre-eclampsia (Sultana et al., 2018) and maternal obesity (Lopez-Jaramillo et al., 2018). Metformin is a commonly used drug during pregnancy (Tarry-Adkins et al., 2021), which is known in other contexts to have anti-ageing effects (Kulkarni et al., 2020). Thus, we investigated whether metformin affects the rate of placental ageing.

We found no evidence of differences in placental ageing trajectory following exposure to metformin either *in vivo* or *in vitro* in humans or mice. Specifically, in placentas from women living with obesity randomized to metformin or placebo during pregnancy, there was no evidence that metformin altered either the placental transcriptome or the progression of gestation-associated changes in methylation. Furthermore, placental telomere length, while decreasing with gestational age, was not different between metformin and placebo-exposed placentas. There was also no histological evidence

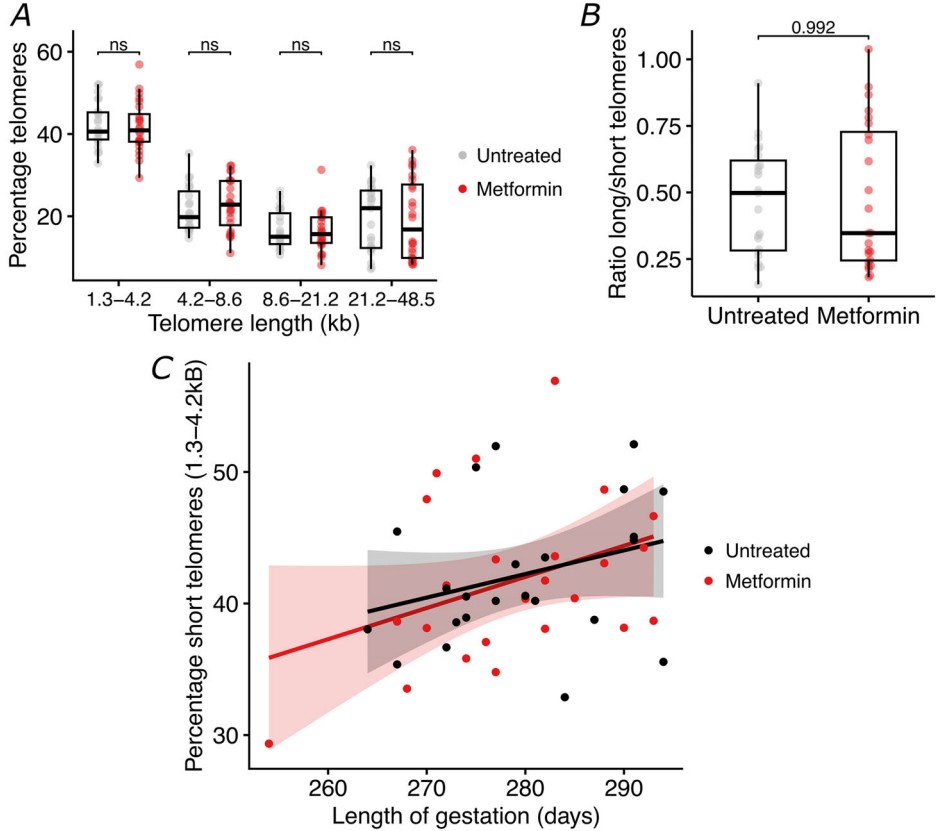

**Figure 5. Telomere length in the placentas of women treated with metformin (*n* = 24) or placebo (*n* = 23) during pregnancy**
*A*, average telomere lengths by metformin treatment status. Medians and IQR are shown. *B*, ratio of long to short telomeres by metformin treatment status. Medians and IQR are shown. *C*, the percentage of short telomeres with length 1.3–2.4 kb over gestation in metformin (red) *versus* control obese (black) placentas. Pairwise comparisons between untreated and metformin-treated placentas were performed using a Mann–Whitney *U* test. [Colour figure can be viewed at wileyonlinelibrary.com]

of altered placental ageing trajectory in placentas exposed to metformin compared to controls. All women included in the EMPOWaR clinical trial had a BMI > 30 kg/m$^2$, and hence these pregnancies were at risk of pathology associated with accelerated placental ageing (Lopez-Jaramillo et al., 2018). We further corroborated our findings in alternative models. Obese mice treated before and during pregnancy with a clinically relevant dose of metformin did not show transcriptomic evidence of advanced placental ageing. Importantly, in this model metformin treatment was initiated before conception, which suggests that the absence of accelerated ageing in the human model is not attributable to late initiation of metformin treatment. Similarly, isolated human trophoblasts treated *in vitro* with metformin did not show reduced expression of ageing-associated genes. Thus, we conclude that there is no evidence that metformin would be an effective treatment to reduce pregnancy complications associated with placental ageing.

Conversely, we found no evidence to suggest that metformin accelerates placental ageing, which supports its continued widespread use in the obstetric setting. Metformin remains an important first-line drug therapy for GDM, which affects one in five pregnant women globally and up to 30–40% in some populations (International Diabetes Federation, 2021). However, of note, an association between metformin use and adverse pregnancy outcomes has been reported (Paschou et al., 2023), although the data remain scarce. Further work is required to investigate other mechanisms through which metformin may influence placental physiology.

Our study has several important strengths, including the use of a range of complementary models to examine the impact of metformin treatment. Our results have direct clinical applicability regarding the use of metformin in the context of pregnancies affected by obesity, an important high-risk group for adverse outcomes of accelerated placental ageing (Catalano & Shankar, 2017). While clinical trials give an important insight into the probable impact of metformin treatment in the clinical setting, interpretation of the results is complicated by potential non-compliance, variation in the timing of dosing and sampling, and variability between subjects. We have therefore complemented this approach with an *in vitro*-treated trophoblast model, which has the important advantage of precise metformin dosing of paired trophoblast samples derived from the same placenta. A further strength of our study is the use of a complementary mouse model that replicates the essential elements of the clinical trial protocol but also addresses the effects of metformin across the whole of gestation.

Limitations of our study design include the lack of information on the impact of metformin specifically during the first trimester. Human placental samples used in our study were treated either *in vivo* from 12 weeks onwards or *in vitro* after delivery. We aim to address the potential benefits of metformin therapy in early pregnancy with future work using placental organoids.

Based on our previous findings of reduced placental energy production via inhibition of complex I in metformin-treated trophoblasts (Tarry-Adkins et al., 2022), we hypothesized that the resultant reduction in oxidative stress markers might slow placental ageing. We speculated that metformin might thus be a potential therapy to prevent pregnancy complications with an aetiological component related to placental ageing. However, we show minimal evidence that metformin affects placental ageing pathways in any of the complementary models that we investigated. Given the high global burden of adverse pregnancy outcomes, including preterm birth, stillbirth and severe fetal growth restriction, further work to find appropriate preventative drug therapies is urgently needed.

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

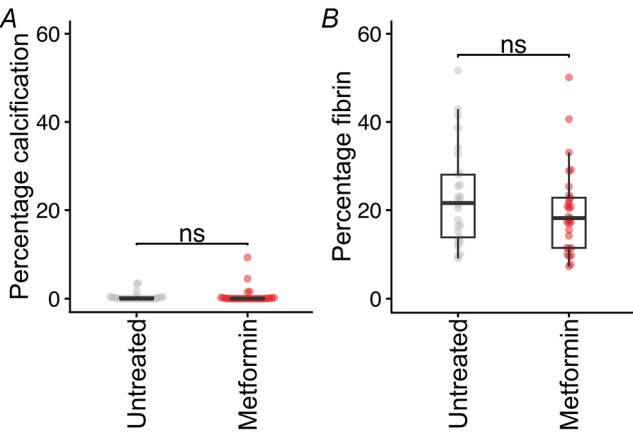

**Figure 6. Placental histology in placentas from women treated with metformin (*n* = 24) or placebo (*n* = 24) during pregnancy**
*A*, percentage placental calcification by metformin treatment status. *B*, percentage placental fibrin by metformin treatment status. Medians and IQR are shown. Unadjusted pairwise comparisons were performed using a Mann–Whitney *U* test. [Colour figure can be viewed at wileyonlinelibrary.com]

Casella, G., Munk, R., Kim, K. M., Piao, Y., De, S., Abdelmohsen, K., & Gorospe, M. (2019). Transcriptome signature of cellular senescence. *Nucleic Acids Research*, **47**(14), 7294–7305.

Catalano, P. M., & Ehrenberg, H. M. (2006). The short- and long-term implications of maternal obesity on the mother and her offspring. *British Journal of Obstetrics and Gynaecology*, **113**(10), 1126–1133.

Catalano, P. M., & Shankar, K. (2017). State of the Art Review: Obesity and pregnancy: Mechanisms of short term and long term adverse consequences for mother and child. *The British Medical Journal*, j1. https://doi.org/10.1136/BMJ.J1

Chawanpaiboon, S., Vogel, J. P., Moller, A. B., Lumbiganon, P., Petzold, M., Hogan, D., Landoulsi, S., Jampathong, N., Kongwattanakul, K., Laopaiboon, M., Lewis, C., Rattanakanokchai, S., Teng, D. N., Thinkhamrop, J., Watananirun, K., Zhang, J., Zhou, W., & Gülmezoglu, A. M. (2019). Global, regional, and national estimates of levels of preterm birth in 2014: A systematic review and modelling analysis. *The Lancet Global Health*, **7**(1), e37–e46.

Chiswick, C., Reynolds, R. M., Denison, F., Drake, A. J., Forbes, S., Newby, D. E., Walker, B. R., Quenby, S., Wray, S., Weeks, A., Lashen, H., Rodriguez, A., Murray, G., Whyte, S., & Norman, J. E. (2015). Effect of metformin on maternal and fetal outcomes in obese pregnant women (EMPOWaR): A randomised, double-blind, placebo-controlled trial. *The Lancet Diabetes & Endocrinology*, **3**(10), 778–786.

Chiswick, C. A., Reynolds, R. M., Denison, F. C., Drake, A. J., Forbes, S., Newby, D. E., Walker, B. R., Quenby, S., Wray, S., Weeks, A., Lashen, H., Rodriguez, A., Murray, G. D., Whyte, S., Andrew, R., Homer, N., Semple, S., Gray, C., Aldhous, M. C., … Norman, J. E. (2016). Does metformin reduce excess birthweight in offspring of obese pregnant women? A randomised controlled trial of efficacy, exploration of mechanisms and evaluation of other pregnancy complications. *Efficacy and Mechanism Evaluation*, **3**(7), 1–800.

Chiswick, C. A., Reynolds, R. M., Denison, F. C., Whyte, S. A., Drake, A. J., Newby, D. E., Walker, B. R., Forbes, S., Murray, G. D., Quenby, S., Wray, S., & Norman, J. E. (2015). Efficacy of metformin in pregnant obese women: A randomised controlled trial. *British Medical Journal Open*, **5**(1), e006854.

Chuprin, A., Gal, H., Biron-Shental, T., Biran, A., Amiel, A., Rozenblatt, S., & Krizhanovsky, V. (2013). Cell fusion induced by ERVWE1 or measles virus causes cellular senescence. *Genes & Development*, **27**(21), 2356–2366.

Cox, L. S., & Redman, C. (2017). The role of cellular senescence in ageing of the placenta. *Placenta*, **52**, 139–145.

Fernandez-Twinn, D. S., Blackmore, H. L., Siggens, L., Giussani, D. A., Cross, C. M., Foo, R., & Ozanne, S. E. (2012). The programming of cardiac hypertrophy in the offspring by maternal obesity is associated with hyper-insulinemia, AKT, ERK, and mTOR activation. *Endocrinology*, **153**(12), 5961–5971.

Ferrari, F., Facchinetti, F., Saade, G., & Menon, R. (2016). Placental telomere shortening in stillbirth: A sign of premature senescence? *The Journal of Maternal-Fetal & Neonatal Medicine*, **29**(8), 1283–1288.

Grundy, D. (2015). Principles and standards for reporting animal experiments in The Journal of Physiology and Experimental Physiology. *The Journal of Physiology*, **593**(12), 2547–2549.

Hug, L., You, D., Blencowe, H., Mishra, A., Wang, Z., Fix, M. J., Wakefield, J., Moran, A. C., Gaigbe-Togbe, V., Suzuki, E., Blau, D. M., Cousens, S., Creanga, A., Croft, T., Hill, K., Joseph, K. S., Maswime, S., Mcclure, E. M., Pattinson, R., … Alkema, L. (2021). Global, regional, and national estimates and trends in stillbirths from 2000 to 2019: A systematic assessment. *The Lancet*, **398**(10302), 772–785.

International Diabetes Federation. (2021). *IDF Diabetes Atlas*: 10th edition. Accessed February 29, 2024. https://diabetesatlas.org/data/en/indicators/14/

Khan, J., Pernicova, I., Nisar, K., & Korbonits, M. (2023). Mechanisms of ageing: Growth hormone, dietary restriction, and metformin. *The Lancet Diabetes & Endocrinology*, **11**(4), 261–281.

Kulkarni, A. S., Gubbi, S., & Barzilai, N. (2020). Benefits of metformin in attenuating the hallmarks of aging. *Cell Metabolism*, **32**(1), 15–30.

Lee, Y., Choufani, S., Weksberg, R., Wilson, S. L., Yuan, V., Burt, A., Marsit, C., Lu, A. T., Ritz, B., Bohlin, J., Gjessing, H. K., Harris, J. R., Magnus, P., Binder, A. M., Robinson, W. P., Jugessur, A., & Horvath, S. (2019). Placental epigenetic clocks: Estimating gestational age using placental DNA methylation levels. *Aging (Albany NY)*, **11**(12), 4238–4253.

Lopez-Jaramillo, P., Barajas, J., Rueda-Quijano, S. M., Lopez-Lopez, C., & Felix, C. (2018). Obesity and pre-eclampsia: Common pathophysiological mechanisms. *Frontiers in Physiology*, **9**, 1838.

Lu, L., Kingdom, J., Burton, G. J., & Cindrova-Davies, T. (2017). Placental stem villus arterial remodeling associated with reduced hydrogen sulfide synthesis contributes to Human fetal growth restriction. *American Journal of Pathology*, **187**(4), 908–920.

Maiti, K., Sultana, Z., Aitken, R. J., Morris, J., Park, F., Andrew, B., Riley, S. C., & Smith, R. (2017). Evidence that fetal death is associated with placental aging. *American Journal of Obstetrics and Gynecology*, **217**(4), 441.e1-441.e14.

Martens, D. S., Plusquin, M., Gyselaers, W., De Vivo, I., & Nawrot, T. S. (2016). Maternal pre-pregnancy body mass index and newborn telomere length. *BioMed Central Medicine [Electronic Resource]*, **14**(1). https://doi.org/10.1186/S12916-016-0689-0

Mayne, B. T., Leemaqz, S. Y., Smith, A. K., Breen, J., Roberts, C. T., & Bianco-Miotto, T. (2017). Accelerated placental aging in early onset preeclampsia pregnancies identified by DNA methylation. *Epigenomics*, **9**(3), 279–289.

Ohmaru-Nakanishi, T., Asanoma, K., Fujikawa, M., Fujita, Y., Yagi, H., Onoyama, I., Hidaka, N., Sonoda, K., & Kato, K. (2018). Fibrosis in preeclamptic placentas is associated with stromal fibroblasts activated by the transforming growth factor-$\beta$1 signaling pathway. *American Journal of Pathology*, **188**(3), 683–695.

Paschou, S. A., Shalit, A., Gerontiti, E., Athanasiadou, K. I., Kalampokas, T., Psaltopoulou, T., Lambrinoudaki, I., Anastasiou, E., Wolffenbuttel, B. H. R., & Goulis, D. G. (2023). Efficacy and safety of metformin during pregnancy: An update. *Endocrine*, **83**(2), 259–269.

Poggi, S. H., Bostrom, K. I., Demer, L. L., Skinner, H. C., & Koos, B. J. (2001). Placental calcification: A metastatic process? *Placenta*, **22**(6), 591–596.

Salomäki, H., Heinäniemi, M., Vähätalo, L. H., Ailanen, L., Eerola, K., Ruohonen, S. T., Pesonen, U., & Koulu, M. (2014). Prenatal metformin exposure in a maternal high fat diet mouse model alters the transcriptome and modifies the metabolic responses of the offspring. *PLoS ONE*, **9**(12), e115778.

Stock, S. J., & Aiken, C. E. (2023). Barriers to progress in pregnancy research: How can we break through? *Science (1979)*, **380**(6641), 150–153.

Sultana, Z., Maiti, K., Dedman, L., & Smith, R. (2018). Is there a role for placental senescence in the genesis of obstetric complications and fetal growth restriction? *American Journal of Obstetrics and Gynecology*, **218**(2), S762–S773.

Tao, Y., Chen, W., Xu, H., Xu, J., Yang, H., Luo, X., Chen, M., He, J., Bai, Y., & Qi, H. (2023). Adipocyte-derived exosomal NOX4-mediated oxidative damage induces premature placental senescence in obese pregnancy. *International Journal of Nanomedicine*, **18**, 4705–4726.

Tarry-Adkins, J. L., Chen, J. H., Smith, N. S., Jones, R. H., Cherif, H., Ozanne, S. E. (2009). Poor maternal nutrition followed by accelerated postnatal growth leads to telomere shortening and increased markers of cell senescence in rat islets. *Federation of American Societies for Experimental Biology Journal*, **23**(5), 1521–1528.

Tarry-Adkins, J. L., Ozanne, S. E., & Aiken, C. E. (2021). Impact of metformin treatment during pregnancy on maternal outcomes: A systematic review/meta-analysis. *Scientific Reports*, **11**(1), 1–13.

Tarry-Adkins, J. L., Ozanne, S. E., Norden, A., Cherif, H., & Hales, C. N. (2006). Lower antioxidant capacity and elevated p53 and p21 may be a link between gender disparity in renal telomere shortening, albuminuria, and longevity. *American Journal of Physiology-Renal Physiology*, **290**(2), F509–F516.

Tarry-Adkins, J. L., Robinson, I. G., Pantaleão, L. C., Armstrong, J. L., Thackray, B. D., Holzner, L. M. W., Knapton, A. E., Virtue, S., Jenkins, B., Koulman, A., Murray, A. J., Ozanne, S. E., & Aiken, C. E. (2023). The metabolic response of human trophoblasts derived from term placentas to metformin. *Diabetologia*, **66**(12), 2320–2331.

Tarry-Adkins, J. L., Robinson, I. G., Reynolds, R. M., Aye, I., Charnock-Jones, D. S., Jenkins, B., Koulmann, A., Ozanne, S. E., & Aiken, C. E. (2022). Impact of metformin treatment on Human placental energy production and oxidative stress. *Frontiers in Cell and Developmental Biology*, **10**. https://doi.org/10.3389/FCELL.2022.935403

Triche, T. J., Weisenberger, D. J., Van Den Berg, D., Laird, P. W., & Siegmund, K. D. (2013). Low-level processing of Illumina Infinium DNA methylation BeadArrays. *Nucleic Acids Research*, **41**(7), e90.

## Additional information

### Data availability statement

The RNAseq data have been deposited on the GEO repository under accession number GSE293162. The gene expression array data are available on the BioStudies database under accession number E-MTAB-6418. The methylation array data can also be accessed on the BioStudies database under accession number E-MTAB-6421.

### Competing interests

None declared.

### Author contributions

G.J.H. analysed the data, prepared figures and drafted the manuscript. L.Y., J.L.T., A.H., K.K.W., D.S.F. and I.G.R. performed experiments, analysed data and edited the manuscript. A.J.D. and S.E.O. conceptualized the study, obtained funding and edited the manuscript. C.E.A. and R.M.R. conceptualized the study, obtained funding, analysed data and drafted the manuscript. C.E.A. is the guarantor of this work and, as such, had full access to all the data in the study and takes responsibility for the integrity of the data and the accuracy of the data analysis.

### Funding

This research was funded by a Medical Research Council New Investigator Grant (MR/T016701/1) to C.E.A. C.E.A. is supported by the NIHR Cambridge Biomedical Research Centre (146281). The views expressed are those of the author(s) and not necessarily those of the NIHR or the Department of Health and Social Care. S.E.O. is supported by the Medical Research Council (MC_UU_00014/4) and the British Heart Foundation (RG/17/12/33167 and PG/20/11/34957). R.M.R. is supported by the Medical Research Council (MR/R014167/1) and the British Heart Foundation (RE/18/5/34216). This work was supported by the Medical Research Council (MC_UU_00039 and MC_UU_00014) and by the Wellcome Trust (226800/Z/22/Z & 208363/Z/17/Z).

### Acknowledgements

We would like to acknowledge the staff at the Institute of Metabolic Science Disease Model Core (DMC) & Histopathology Core Facility for histological analysis and the Genomics & Bioinformatics Core Facility for RNAseq analyses.

### Keywords

fetus, gestational diabetes, metformin, placenta, pregnancy, trophoblast

### Supporting information

Additional supporting information can be found online in the Supporting Information section at the end of the HTML view of the article. Supporting information files available:

**Peer Review History**

