## [Peer Review History · The Journal of Physiology]

The impact of metformin on placental ageing in humans and mice

Grace Jennifer Hattersley, Liu Yang, Jane L Tarry Adkins, Antonia Hufnagel, Kwun Kiu Wong, Denise S Fernandez-Twinn, India G Robinson, Maria Chukanova, Amanda J Drake, Rebecca Reynolds, Susan E Ozanne, and Catherine Aiken
DOI: 10.1113/JP288710

Corresponding author(s): Catherine Aiken (cema2@cam.ac.uk)

The following individual(s) involved in review of this submission have agreed to reveal their identity: Thomas Jansson (Referee #1)

Review Timeline:

Submission Date:	09-Feb-2025
Editorial Decision:	19-Mar-2025
Revision Received:	24-Apr-2025
Editorial Decision:	01-May-2025
Revision Received:	12-May-2025
Accepted:	14-May-2025

Senior Editor: Laura Bennet

Reviewing Editor: Rebecca Simmons

Transaction Report:

Dear Dr Aiken,

Re: JP-RP-2025-288710 "**The impact of metformin on placental ageing in humans and mice**" by Grace Jennifer Hattersley, Liu Yang, Jane L Tarry Adkins, Antonia Hufnagel, Kwun Kiu Wong, Denise S Fernandez-Twinn, India G Robinson, Amanda J Drake, Rebecca Reynolds, Susan E Ozanne, and Catherine E Aiken

Thank you for submitting your manuscript to The Journal of Physiology. It has been assessed by a Reviewing Editor and by 2 expert referees and we are pleased to tell you that it is acceptable for publication following satisfactory revision.

REVISION CHECKLIST:

We look forward to receiving your revised submission.

Yours sincerely,

Laura Bennet
Senior Editor
The Journal of Physiology

REQUIRED ITEMS

- Author photo and profile. First or joint first authors are asked to provide a short biography (no more than 100 words for one author or 150 words in total for joint first authors) and a portrait photograph. These should be uploaded and clearly labelled together in a Word document with the revised version of the manuscript. See Information for Authors for further details.

- You must start the Methods section with a paragraph headed Ethical Approval. If experiments were conducted on humans, confirmation that informed consent was obtained, preferably in writing, that the studies conformed to the standards set by the latest revision of the Declaration of Helsinki and that the procedures were approved by a properly constituted ethics committee, which should be named, must be included in the article file. If the research study was registered (clause 35 of the Declaration of Helsinki), the registration database should be indicated, otherwise the lack of registration should be noted as an exception (e.g. The study conformed to the standards set by the Declaration of Helsinki, except for registration in a database). For further information see: <https://physoc.onlinelibrary.wiley.com/hub/human-experiments>.

- The reference list must be in alphabetical order, rather than numbered, to comply with our Journal format.

- Please upload separate high-quality figure files via the submission form.

- Please ensure that any tables are editable and in Word format, and wherever possible, embedded in the article file itself.

- Please ensure that the Article File you upload is a Word file.

- Please include an Abstract Figure file, as well as the Figure Legend text within the main article file. The Abstract Figure is a piece of artwork designed to give readers an immediate understanding of the research and should summarise the main conclusions. If possible, the image should be easily 'readable' from left to right or top to bottom. It should show the physiological relevance of the manuscript so readers can assess the importance and content of its findings. Abstract Figures should not merely recapitulate other figures in the manuscript. Please try to keep the diagram as simple as possible and without superfluous information that may distract from the main conclusion(s). Abstract Figures must be provided by authors no later than the revised manuscript stage and should be uploaded as a separate file during online submission labelled as File Type 'Abstract Figure'. Please also ensure that you include the figure legend in the main article file. All Abstract Figures should be created using BioRender. Authors should use The Journal's premium BioRender account to export high-resolution images. Details on how to use and access the premium account are included as part of this email.

EDITOR COMMENTS

Reviewing Editor:

Thank you for submitting your paper to our journal. The reviewers were overall very positive about your study and the issues they raised are important, but should be easily addressable.

REFeree COMMENTS

Referee #1:

In this paper, Hattersley and coworkers tested the hypothesis that metformin treatment in pregnancy in women and mice alter placental ageing. Results were validated in cultured primary human trophoblast cells. The premise for the study is previous reports showing that metformin may slow cellular ageing by improving nutrient-sensing, enhancing autophagy, protecting against macromolecular damage, delaying stem-cell aging, modulating mitochondrial function, regulating transcription, and lowering telomere attrition and senescence. In the current study, however, there were no evidence that metformin affects placental ageing pathways. The strength of the study is the use of multiple approaches to assess ageing in a large cohort of placentas from women treated with metformin. The paper is well-written, methods used are rigorous and the report provides some new important information, despite the 'negative' findings.

Some minor comments:

1. Key points line 26-27: Conclusions "Lack of accelerated ageing is reassuring with respect to treatment strategies for gestational diabetes'. This statement makes little sense given that (1) metformin is known to decrease cellular ageing so accelerated ageing was not expected and (2) metformin may have a range of negative effects on the placenta and fetus not studied in this report so not clear how reassuring the finding of no effect on placental ageing is for the use of metformin in pregnancy?
2. Please provide information on hCG secretion from cultured PHT cells to confirm that hCG secretion is not in decline at 120 hours when cells were harvested.
3. Discussion Line 395-397: The sentence "More work is urgently needed to find drug therapies that are potential candidates to slow accelerated placental ageing, particularly in view of the high global burden of associated complications including preterm birth, stillbirth, and severe fetal growth restriction" may be an overstatement given that no cause-and-effect relationship between placental ageing and these pregnancy complications has been demonstrated.

Referee #2:

The premise of this study is based on prior observations that metformin may have anti-aging effects. Since metformin is used by many women during pregnancy and because accelerated placental "aging" is associated with adverse pregnancy outcomes, they sought to evaluate if metformin could provide benefit to slow placental aging.

Strengths of the manuscript include the use of sophisticated and comprehensive methods to assess placental aging with telomere light, RNA-seq, and DNA methylation. They also used placental samples from the EMPOWAR cohort along with isolated trophoblasts and samples from a mouse model. Robust statistical measures with FDR adjusted rates were used.

Overall, this manuscript answers an important question in the field regarding the beneficial or harmful effects of metformin use during pregnancy. The results demonstrate that metformin had no added benefit on slowing placental aging and that metformin had no added harm on accelerated aging. Both are important outcomes.

The following are concerns about the manuscript:

1. The highlights should be revised - repeat of the abstract and may need to follow journal guidelines
2. The abstract could be revised to include more detail. It is over simplistic and fails to capture the depth of the study and the importance of the question being asked (opening sentences) and implications (benefit vs harm).
3. Placental aging could be better defined. Not immediately clear if this refers to the gestational age of the placenta or maternal age effects. Also, does aging in the context of adverse pregnancies mean that that placental function is simply more mature? If not, is aging the correct term? Sounds more like placental dysfunction.
4. The human cohort only include women with BMI >30 who were taking metformin or placebo. - con group?? Can you comment as to how the placebo group compares to normal weight placentas?
5. Considering expanding the conclusions to state that metformin in setting of maternal obesity doesn't worsen nor improve placental aging readouts. Both are important for the field. This is well addressed in the Discussion but could be added to the

Abstract and Highlights.

6. Human gene symbols should be all capitalized and italics.

END OF COMMENTS

The impact of metformin on placental ageing in humans and mice
Response to referees

Dear Prof. Bennet,

We are grateful to yourself, the reviewing editor and the expert referees for the careful consideration of our manuscript "*The impact of metformin on placental ageing in humans and mice*" for publication at The Journal of Physiology. We have addressed all the reviewers' comments provided and enclose our point-by-point responses below. We have also provided the updated manuscript, including a tracked and clean version. Of note, Dr Maria Chukanova was instrumental in helping us to address Reviewer 1's second the comment on hCG secretion from cultured primary human trophoblast cells and thus has been added to the author list.

Many thanks,

Dr Catherine Aiken, MB/BChir, MA, PhD, MRCOG, MRCP
Professor of Maternal and Fetal Medicine
School of Clinical Medicine, University of Cambridge, UK

Reviewer 1

We thank the reviewer for their thoughtful consideration of our manuscript and helpful comments, which have helped to strengthen our study. Please see our point-by-point responses as follows.

- 1) Key points line 26-27: Conclusions "Lack of accelerated ageing is reassuring with respect to treatment strategies for gestational diabetes'. This statement makes little sense given that (1) metformin is known to decrease cellular ageing so accelerated ageing was not expected and (2) metformin may have a range of negative effects on the placenta and fetus not studied in this report so not clear how reassuring the finding of no effect on placental ageing is for the use of metformin in pregnancy?

The reviewer raises good points. We did not hypothesise that metformin would accelerate cellular ageing in the placenta, which as the reviewer points out, would be completely unexpected in the context of the present literature. However, we believe it is worth noting that our data does not indicate any unexpected acceleration of placental ageing, which would seriously challenge the safety of the use of metformin in pregnancy. We further note R2's comment 5 to further highlight the absence of accelerated placental ageing.

Indeed, others have reported potentially adverse outcomes associated with metformin use (Paschou et al., 2023). Importantly, our data would suggest that this association is mediated by mechanisms independent of placental ageing. We have now amended the key points and discussion to reflect the reviewer's points.

- 2) Please provide information on hCG secretion from cultured PHT cells to confirm that hCG secretion is not in decline at 120 hours when cells were harvested.

Thank you for your helpful prompt to include this data. We have now added representative data showing ongoing beta-hCG increases throughout the culture period in supplementary figure 2.

- 3) Discussion Line 395-397: The sentence "More work is urgently needed to find drug therapies that are potential candidates to slow accelerated placental ageing, particularly in view of the high global burden of associated complications including preterm birth, stillbirth, and severe fetal growth restriction" may be an overstatement given that no cause-and-effect relationship between placental ageing and these pregnancy complications has been demonstrated.

The text has been updated to reflect the need to find drug therapies to prevent these pregnancy complications through any mechanism, regardless of relation to placental ageing.

Reviewer 2

We are grateful to the reviewer for their careful evaluation of our manuscript and constructive comments, which have helped to strengthen our study. Please see our point-by-point responses as follows.

- 1) The highlights should be revised - repeat of the abstract and may need to follow journal guidelines

We have updated this section accordingly to better fit journal guidelines and to provide a different tone from the abstract.

- 2) The abstract could be revised to include more detail. It is over simplistic and fails to capture the depth of the study and the importance of the question being asked (opening sentences) and implications (benefit vs harm).

The abstract background has been updated to further reflect the context and importance of our study and to provide further supporting information on our hypothesis that metformin could slow placental ageing. We have also addressed the implications further, providing additional information on the consequences of metformin not affecting placental ageing.

- 3) Placental aging could be better defined. Not immediately clear if this refers to the gestational age of the placenta or maternal age effects. Also, does aging in the context of adverse pregnancies mean that that placental function is simply more mature? If not, is aging the correct term? Sounds more like placental dysfunction.

Placental ageing has been further defined in the abstract to help distinguish this from gestational age and maternal ageing.

- 4) The human cohort only include women with BMI >30 who were taking metformin or placebo. - con group?? Can you comment as to how the placebo group compares to normal weight placentas?

The reviewer raises an interesting point that placental physiology differs between normal weight and obese placentas (Howell & Powell, 2017; Kelly *et al.*, 2023; Santos *et al.*, 2023) therefore any effect of metformin may be different in these two groups. However, this was outside of the scope of the EMPOWaR trial (*Methods*), which this study is based on. Moreover, it would be clinically and ethically inappropriate to administer metformin to otherwise healthy women, in whom treatment is not indicated, given the potential side effects (Chiswick *et al.*, 2015).

- 5) Considering expanding the conclusions to state that metformin in setting of maternal obesity doesn't worsen nor improve placental aging readouts. Both are important for the field. This is well addressed in the Discussion but could be added to the Abstract and Highlights.

We have now added this point to the abstract and key points. However, we have remained cautious in commenting on how reassuring this finding is, because as R1 points out, other mechanisms may influence outcomes independent of placental ageing.

- 6) Human gene symbols should be all capitalized and italics.

All gene symbols have been reviewed and updated as advised.

Dear Dr Aiken,

Re: JP-RP-2025-288710R1 "**The impact of metformin on placental ageing in humans and mice**" by Grace Jennifer Hattersley, Liu Yang, Jane L Tarry Adkins, Antonia Hufnagel, Kwun Kiu Wong, Denise S Fernandez-Twinn, India G Robinson, Maria Chukanova, Amanda J Drake, Rebecca Reynolds, Susan E Ozanne, and Catherine Aiken

I am pleased to let you know that your manuscript is almost ready for acceptance. Before formal acceptance, however, we would be grateful if you could include your supplemental material within the article itself, to comply with our Supporting Information policy (https://jp.msubmit.net/cgi-bin/main.plex?form_type=display_requirements#suppinfo).

REVISION CHECKLIST:

We look forward to receiving your revised submission.

Yours sincerely,

Laura Bennet
Senior Editor
The Journal of Physiology

EDITOR COMMENTS

Reviewing Editor:

Thank you for submitting your manuscript and for addressing all of the reviewers' comments

REFEREE COMMENTS

Referee #1:

The responses to the reviewers' comments are satisfactory. It is suggested that authors provide a unit for hCG secretion in Fig 1C: for example is it total secretion/24 hours or a concentration (hCG/ml/24 hours) or something else? Also, adding a line to a few data points originating from one cell preparation seems unwarranted and the line should be removed.

END OF COMMENTS

The impact of metformin on placental ageing in humans and mice

Response to referees

Reviewer 1

The responses to the reviewers' comments are satisfactory. It is suggested that authors provide a unit for hCG secretion in Fig 1C: for example is it total secretion/24 hours or a concentration (hCG/ml/24 hours) or something else? Also, adding a line to a few data points originating from one cell preparation seems unwarranted and the line should be removed.

Thank you for reviewing our updated submission. The figure has been edited to remove the line and to indicate the unit of secretion (IU/L).

Dear Dr Aiken,

Re: JP-RP-2025-288710R2 "**The impact of metformin on placental ageing in humans and mice**" by Grace Jennifer Hattersley, Liu Yang, Jane L Tarry Adkins, Antonia Hufnagel, Kwun Kiu Wong, Denise S Fernandez-Twinn, India G Robinson, Maria Chukanova, Amanda J Drake, Rebecca Reynolds, Susan E Ozanne, and Catherine Aiken

We are pleased to tell you that your paper has been accepted for publication in The Journal of Physiology.

Yours sincerely,

Laura Bennet
Senior Editor
The Journal of Physiology

If you would like to receive our 'Research Roundup', a monthly newsletter highlighting the cutting-edge research published in The Physiological Society's family of journals (The Journal of Physiology, Experimental Physiology, Physiological Reports, The Journal of Nutritional Physiology and The Journal of Precision Medicine: Health and Disease), please click this link, fill in your name and email address and select 'Research Roundup':
<https://www.physoc.org/journals-and-media/membernews>

- **TRANSPARENT PEER REVIEW POLICY:** To improve the transparency of its peer review process, The Journal of Physiology publishes online as supporting information the peer review history of all articles accepted for publication. Readers will have access to decision letters, including Editors' comments and referee reports, for each version of the manuscript as well as any author responses to peer review comments. Referees can decide whether or not they wish to be named on the peer review history document.
- You can help your research get the attention it deserves! Check out Wiley's free Promotion Guide for best-practice recommendations for promoting your work at: www.wileyauthors.com/eo/guide. You can learn more about Wiley Editing Services which offers professional video, design, and writing services to create shareable video abstracts, infographics, conference posters, lay summaries, and research news stories for your research at: www.wileyauthors.com/eo/promotion.
- **IMPORTANT NOTICE ABOUT OPEN ACCESS:** To assist authors whose funding agencies mandate public access to published research findings sooner than 12 months after publication, The Journal of Physiology allows authors to pay an Open Access (OA) fee to have their papers made freely available immediately on publication.
